# Machine learning driven dashboard for chronic myeloid leukemia prediction using protein sequences

Waqar Ahmad[1], Abdul Raheem Shahzad[2], Muhammad Awais Amin[1,3],
Waqas Haider Bangyal[4], Tahani Jaser Alahmadi[5]*, Saddam Hussain Khan[6]

1 Department of Computer and Information Sciences, Pakistan Institute of Engineering and Applied Sciences, Islamabad, Pakistan, 2 CECOS University of IT and Emerging Sciences, Peshawar, Khyber Pakhtunkhwa (KPK), Pakistan, 3 Data Science Consultant, Datamatics Technologies, Islamabad, Pakistan, 4 Department of Computer Science, Kohsar University Murree, Punjab, Pakistan, 5 Department of Information Systems, College of Computer and Information Sciences, Princess Nourah bint Abdulrahman University, Riyadh, Saudi Arabia„ 6 Artificial Intelligence Lab, Department of Computer Systems Engineering, University of Engineering and Applied Sciences (UEAS), Swat, Pakistan

☢ These authors contributed equally to this work.
* tjalahmadi@pnu.edu.sa

**Data availability statement:** All relevant data for this study are publicly available from the GitHub repository (https://github.com/awaismalik1x/CML_Prediction_Data.git).

## Abstract

The prevalence of Leukaemia, a malignant blood cancer that originates from hematopoietic progenitor cells, is increasing in Southeast Asia, with a worrisome fatality rate of 54%. Predicting outcomes in the early stages is vital for improving the chances of patient recovery. The aim of this research is to enhance early-stage prediction systems in a substantial manner. Using Machine Learning and Data Science, we exploit protein sequential data from commonly altered genes including BCL2, HSP90, PARP, and RB to make predictions for Chronic Myeloid Leukaemia (CML). The methodology we implement is based on the utilisation of reliable methods for extracting features, namely Di-peptide Composition (DPC), Amino Acid Composition (AAC), and Pseudo amino acid composition (Pse-AAC). We also take into consideration the identification and handling of outliers, as well as the validation of feature selection using the Pearson Correlation Coefficient (PCA). Data augmentation guarantees a comprehensive dataset for analysis. By utilising several Machine Learning models such as Support Vector Machine (SVM), XGBoost, Random Forest (RF), K Nearest Neighbour (KNN), Decision Tree (DT), and Logistic Regression (LR), we have achieved accuracy rates ranging from 66% to 94%. These classifiers are thoroughly evaluated utilising performance criteria such as accuracy, sensitivity, specificity, F1-score, and the confusion matrix.The solution we suggest is a user-friendly online application dashboard that can be used for early detection of CML. This tool has significant implications for practitioners and may be used in healthcare institutions and hospitals.

**Funding:** The authors are grateful to the Princess Nourah bint Abdulrahman University Researchers Supporting Project number (PNURSP2024R513) at Princess Nourah bint Abdulrahman University, Riyadh, Saudi Arabia, for providing the necessary funding for this work.

**Competing interests:** The authors have declared that no competing interests exist.

## Introduction

Leukemia is a complex medical condition influenced by genetic regulation in the production of blood cells. When hematopoietic precursor cells turn malignant [1], it gives rise to abnormal cell growth due to alterations in DNA and RNA sequences. This transformation results in the infiltration of healthy cells by malignant ones, thus causing Leukemia. The illness primarily entails the uncontrolled proliferation of specifically White Blood Cells (WBC), i.e., neutrophils, basophils, and eosinophils, while lymphocytes remain unaffected. Acute myeloid Leukemia (AML), chronic myeloid Leukemia (CML), acute lymphoblastic Leukemia (ALL), and chronic lymphocytic Leukemia (CLL) are some of the several kinds of Leukemia [2]. The only subject of our research is Chronic Myeloid Leukemia (CML).

Leukemia cancer presents a substantial health challenge due to the abnormal proliferation of White Blood Cells (WBC) [1]. While research has concentrated on detecting cancer through blood cell images, exploration of Protein Sequential data is limited. Leukemia diagnosis heavily relies on hematologists, posing limitations in regions with a scarcity of specialists. Mortality rates are on the rise, particularly in South East Asia [3], creating a demand for an early detection approach. The motivation for driving the proposed research arises from the observation that a plethora of research has been conducted on cancer predictions—such as lung cancer, liver cancer, colon cancer, ovarian cancer, etc. utilizing MRI (magnetic resonance imaging), CT (computed tomography) scans, image processing techniques and protein sequences [4–6]. However, the realm of gene data in bioinformatics remains relatively uncharted, especially within the context of Chronic Myeloid Leukemia (CML). At present, no AI-based Dashboard system predicts Leukemia based on protein sequences, but developing such a system could revolutionize the diagnosis, leading to saved lives and eased healthcare burdens. Collaborative efforts between Machine Learning and Data Science can establish a robust model for accessible and timely Leukemia solutions.

As illustrated in Fig 1, the proposed research suggests the utilization of Machine Learning-based techniques to identify genes that cause Leukemia through Protein Sequences, aiming for early detection and a reduction in the mortality rate. This undertaking could emerge as a flagship initiative in health sciences, addressing the shortage of specialized hematologists. Implementation of the system would result in timely interventions and improved recovery prospects. Automating certain diagnostic processes could ease the load on specialists and

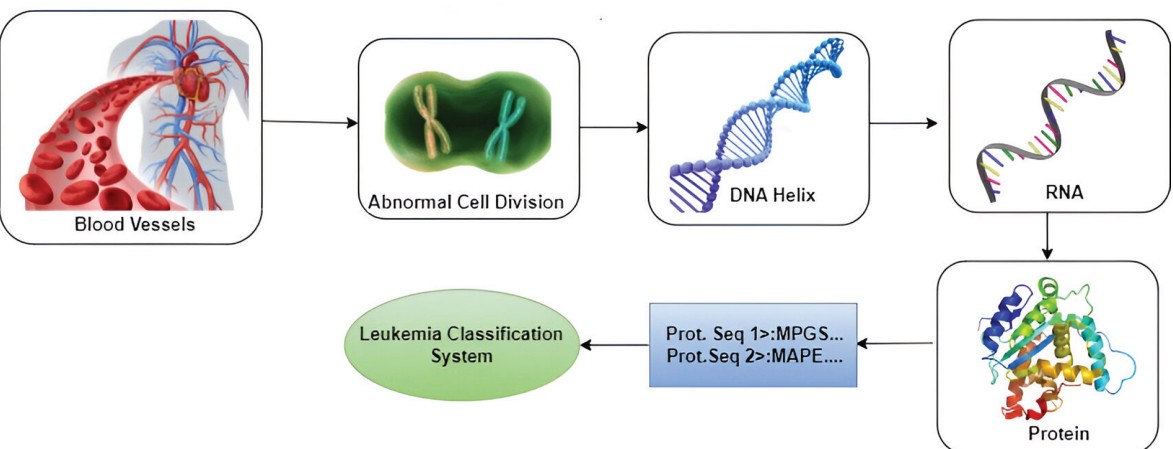

**Fig 1. Various stages of chronic Myeloid leukemia classification.**

enhance healthcare services. The potential impact goes beyond Leukemia diagnosis, garnering recognition, and interest from the medical community. Overall, this AI-driven research holds immense promise in reshaping healthcare and propelling the advancement of AI applications. Because of this research, innovative insights, and progress in predicting and comprehending CML could come to fruition. This might lead to more effective diagnostic and treatment methodologies, benefiting patients and healthcare systems. Furthermore, the successful integration of bioinformatics and AI could pave the way for pioneering applications and further interdisciplinary research at the intersection of these two promising domains.

The main contribution of our proposed research is as follows:

- The current study focuses on protein sequential data rather than image data.
- The most frequently mutated genes that were responsible for chronic myeloid leukemia were discovered through a literature review.
- Datasets were formulated from the most frequently mutated gene data.
- Features were gathered through analysing the physicochemical features of the amino acid composition, pseudo amino acid composition, and di-peptide composition.
- The study aims to increase patient recovery prospects by improved early-stage prognosis.
- The solution we suggest is a user-friendly online application dashboard that serves as a vital tool for early identification of CML. It can be easily implemented in healthcare facilities and hospitals.

This paper follows a structured format that aims to understand the research comprehensively. **Introduction**, outlines the problem statement. **Literature review**, discusses related research, positioning our study in the existing body of knowledge. **Materials and methods**, details the dataset creation process and experimental techniques. **Development of individual classifiers**, presents our methodology and analysis. **Results and discussion**, succinctly interprets the findings. Lastly, we offer a **Conclusion** summarizing our contributions and outlining future research directions.

## Literature review

This section comprehensively discusses the recently conducted Leukemia research, focusing on Protein Sequences, RNA, and blood cell imagery. It elaborates acquiring and forming the dataset, which is pivotal in creating standardized Leukemia datasets by utilizing protein sequences. Importantly, previous researchers have not combined these three distinct feature extraction techniques while implementing a user-friendly dashboard, as done in this study. In [7], the Random Forest model was utilized to diagnose the cancerous growth of White Blood Cells with an accuracy of 94.3%. In the research by [8], the classifier was evaluated using 60 photos, demonstrating that models like K-nearest neighbors and Naive Bayes Classifier could identify ALL with an accuracy of 92.8%. According to research [9], the Artificial Bee Colony algorithm – Back Propagation Neural Network (ABC-BPNN) scheme and Principal Component Analysis (PCA) were used to classify Leukemia cells with an average accuracy of 98.72% while also speeding up the calculation.

In reference [10] Jothi et al. investigated the identification of leukemia sub-types, particularly ALL, using BSA-based clustering and advanced classification algorithms such as decision tree (DT), K-nearest neighbor (KNN), Naive Bayes (NB), and Support Vector Machine (SVM). The SVM model exhibited an accuracy rate of 89.81%. The SVM model was used in research [11] to identify ALL, with an accuracy rate of 89.81%. The dataset was used in [12] to

classify ALL using the K-nearest neighbor method, with a 96.25% accuracy rate. In study gal [36,37], the exploration centered around the use of ML algorithms to analyze gene expression patterns derived from RNA sequencing (RNA-seq) for accurately predicting the likelihood of CR in pediatric AML patients' post-induction therapy. Research [38] Developed models for predicting and classifying different stages of colon cancer using RNA-seq data of extracellular vesicles (EV) from healthy individuals and colon cancer patients. The study employed five canonical ML and Deep Learning (DL) classifiers, achieving high accuracy rates, resulting in an accuracy of 94.6% for K-nearest neighbor, 97.33% for Random Forest, 93% for LMT, and 92% for Random Tree. In [39], the early diagnosis and distinction between types of lung cancers, i.e., Non-Small Cell Lung Cancer and Small Cell Lung Cancer, were highlighted as crucial for improving patient survival rates. The proposed diagnostic system utilized sequence-derived structural and physicochemical attributes of proteins associated with tumor types, employing feature extraction, selection, and prediction models.

The study conducted by Dhakal et al. [40,41] Developed a stacking classifier method that specifically targets CTS selection criteria by utilising feature-encoding approaches. This algorithm generates feature vectors that include k-mer nucleotide composition, dinucleotide composition, pseudo-nucleotide composition, and sequence order coupling. The stacking classifier method demonstrated superior performance compared to prior cutting-edge algorithms in identifying functional miRNA targets, with an accuracy rate of 79.77%. In another study, Albitar et al. [50], Using Next Generation Sequencing (NGS) and targeted RNA sequencing along with a machine learning approach, Albitar et al. investigated the potential of discovering new biomarkers that can predict Acute graft-vs.-host disease (aGVHD). The study by Ahmad et al. [51], Predicted chronic Lymphocytic Leukemia using protein sequences with Chou's Pseudo Amino Acid Composition (PseAAC) and statistical moments. In the study Jian et al.[52] utilised deep learning (DL) to develop a prediction model only for transcription factor binding sites, utilising just the original DNA base sequences. In this study, a deep learning approach utilising convolutional neural network (CNN) and long short-term memory (LSTM) was developed to analyse four distinct categories of Leukaemia based on transcription factor binding sites. The analysis was conducted using four extensive non-redundant datasets for acute, chronic, myeloid, and lymphatic Leukaemia. The method achieved an average prediction accuracy of 75%.

## Materials and methods

The proposed research centers on the detection of leukemia, specifically targeting Chronic Myeloid Leukemia (CML), characterized by the neoplastic proliferation of White Blood Cells (WBCs) such as neutrophils, basophils, and eosinophils, while excluding lymphocytes. As previously mentioned, CML is linked to a heightened mortality rate due to its typical diagnosis at advanced stages, posing challenges for effective recovery. In response to this concern, we aim to create a dashboard to identify leukemia utilizing Protein Sequential data. To achieve this goal, we collected data on the most frequently mutated genes related to leukemia cancer, leveraging the physiochemical properties of protein sequences for feature extraction. Subsequently, data augmentation techniques were applied to enhance the extracted features, while outliers were detected and removed to ensure data quality. We employed a diverse set of machine learning algorithms, including Support Vector Machine (SVM) [14,15,53,57], XG Boost, Random Forest [16,17], KNN [18,19], logistic regression [54,58,59], and decision tree, as comprehensively described in a study review [20,21,26,55].

The accuracy of each algorithm was evaluated, and the one exhibiting the highest accuracy was selected for integration into our system. This chosen algorithm determines the presence

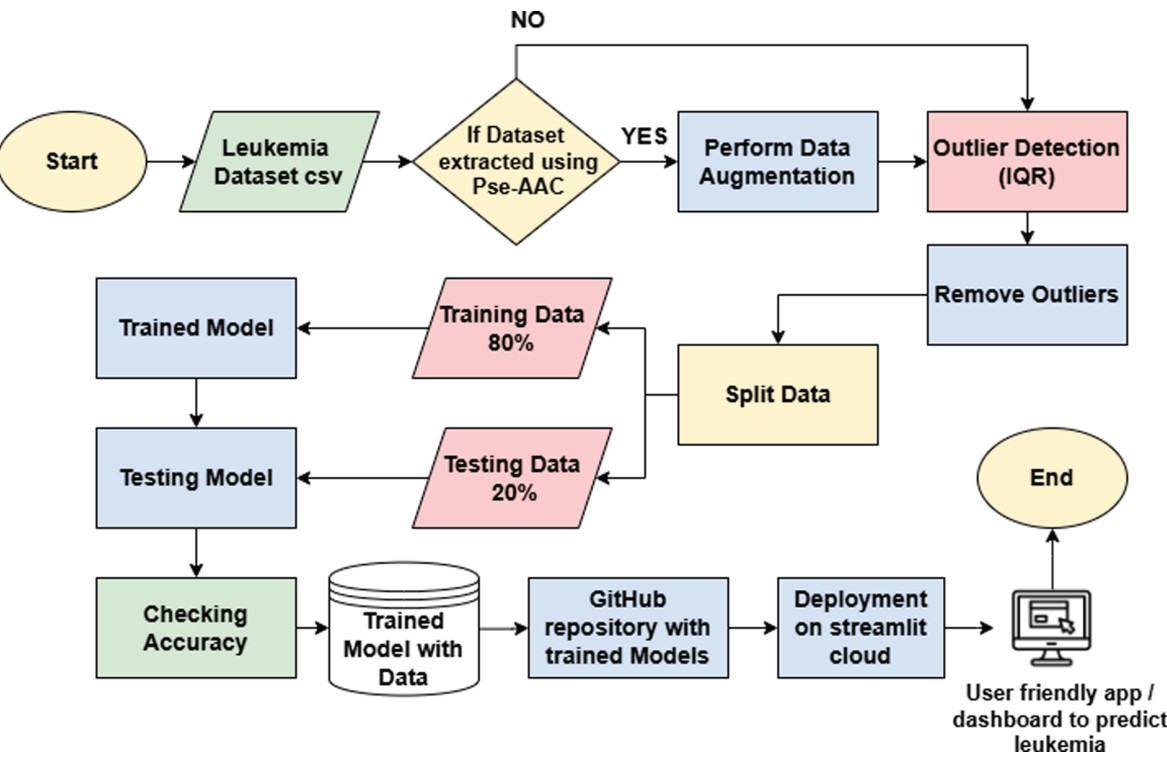

**Fig 2. Block diagram of designed system.**

or absence of cancer in an individual. Finally, we serialized our model using tools such as Pickle or Joblib, facilitating the preservation of the trained model alongside its associated data. These trained models were then incorporated into a Streamlit-based dashboard, enhancing their user-friendly deployment in hospitals and other medical facilities (see Fig 2).

## Block diagram

## Dataset collection

The dataset for this study was collected from the UniProt database, which is a comprehensive resource for protein sequence and functional information. A keyword search was conducted on UniProt using terms such as "Chronic Myeloid Leukemia," "BCL2," "HSP90," "PARP," and "RB." This search yielded a total of 2248 protein sequences. mutated, i.e. BCL2, HSP90, PARP and RB, were utilized for CML [14]. Moreover, the homologous samples were eliminated by maintaining 0.6 as the cutoff level [16]. HSP90 functions as a chaperone protein, crucial in protein folding and degradation processes. Its up-regulation has been identified in various cancer types, including chronic myeloid leukemia (CML). Extensive research has demonstrated that inhibiting HSP90 can attenuate the growth of CML cells and enhance their susceptibility to chemotherapy and tyrosine kinase inhibitors (TKIs) [42,43]. PARP (Poly ADP-ribose polymerase) is an essential enzyme involved in DNA repair processes. Inhibiting PARP has demonstrated effectiveness in the treatment of cancers with BRCA mutations, and there is emerging evidence suggesting its potential applicability in managing chronic myeloid leukemia (CML) [44,45].

The BCL2 (B-cell lymphoma 2) protein family plays a crucial role in regulating programmed cell death, known as apoptosis. Elevated levels of BCL2 have been linked to resistance to chemotherapy in chronic myeloid leukemia (CML) cells. Studies have demonstrated that inhibiting BCL2 can reinstate apoptosis in CML cells and boost the effectiveness of tyrosine kinase inhibitors (TKIs) [46,47]. RB (Retinoblastoma) is a pivotal tumor suppressor gene involved in regulating cell cycle progression. The deactivation of RB is a prevalent characteristic in CML, and research has established that its reactivation can impede the proliferation of CML cells [48,49]. The FASTA file format was used to extract the CML-related protein sequences from the Universal Resource of Proteins (UniProtKB) [15,22]. A successful dataset was created as a result. The same number of negative and positive samples were gathered for CML using the opposite query phrase to create a negative dataset. Consequently, the dataset created for CML is balanced.

**Fasta format.** In bioinformatics, the fasta format is a popular text-based format for representing proteins. It is derived from the FASTA software suite and follows a specific structure. A FASTA sequence starts with a single line that serves as a description and is followed by lines containing the sequencing data [22]. The description line is distinguished from the sequence data by the presence of a greater-than symbol (">") in the first column. The term following the "|" sign is used to identify the sequence, while the rest of the line can be used to provide an additional description, though both are optional.

**Sample of protein sequence (HSP90).** Initially, protein sequences contained redundant data. We employed a benchmark method known as CD-Hit to address the issue of redundant data within the initial protein sequences (see Fig 3). It is essential to utilize a benchmark algorithm for redundancy removal to ensure the validity and reliability of the data. CD-Hit, an online clustered database, was selected for this purpose, with a threshold of 0.6 [23]. This threshold value helps in effectively removing redundancy while preserving the integrity of the dataset.

## Feature extraction

This section elaborates on the feature extraction techniques using physiochemical properties of the protein sequences. These techniques enable the effective representation of protein sequences and extraction of meaningful information crucial for predicting Chronic Myeloid Leukemia. The feature extraction methods utilized in this study fall into three categories:

**Amino acid composition.** The presence of specific amino acids often in a protein sequence is highlighted by AAC characteristics [24,25]. The percentage frequency of an amino acid, $FAAC_{i,j}$, in the $j^{th}$ protein is calculated using the formula below:

$$FAAC_{i,j} = \left(\frac{n_{i,j}}{n_{a,j}}\right) \times 100 \tag{1}$$

```
>sp|Q07817|B2CL1_HUMAN Bcl-2-like protein 1 OS=Homo sapiens OX=9606 GN=BCL2L1 PE=1 SV=1
MSQSNRELVVDFLSYKLSQKGYSWSQFSDVEENRTEAPEGTESEMETPSAINGNPSWHLA
DSPAVNGATGHSSSLDAREVIPMAAVKQALREAGDEFELRYRRAFSDLTSQLHITPGTAY
QSFEQVVNELFRDGVNWGRIVAFFSFGGALCVESVDKEMQVLVSRIAAWMATYLNDHLEP
WIQENGGWDTFVELYGNNAAAESRKGQERFNRWFLTGMTVAGVVLLGSLFSRK
```

**Fig 3. Sample of protein sequence (HSP90).**

In the above equation, $n$ denotes the amount of amino acids type ($i$) found in proteins $j$ while $n_{a,j}$ refers to the total amount of amino acids contained in a protein. The $j^{th}$ protein sequence in the FAAC features dataset is represented as a 20-dimensional (20-D) feature vector as follows:

$$X_j = \left[FAAC_{1,j}, FAAC_{2,j}, \dots, FAAC_{20,j}\right]^T \tag{2}$$

where $X_j = \left[FAAC_{1,j}, FAAC_{2,j}, \dots, FAAC_{20,j}\right]^T$ demonstrates how amino acids are composed.

The technique of amino acid composition involves extracting features from our data, resulting in a 20-dimensional feature set. However, the problem with this approach lies in the limited usefulness of the features extracted. Despite employing various data science feature engineering approaches and conducting hyper-parameter tuning, accuracy remains constrained. Consequently, this approach proves less efficacious in attaining the desired outcomes.

**Pseudo amino acid composition.** A 25-dimensional feature set is produced using the Pseudo Amino Acid Composition (PAAC) approach to extract features from our data [13]. The remarkable fact is that the features extracted through this method are highly valuable. By further applying data science methods and feature engineering techniques, accuracy significantly improves, reaching an impressive range of 91% to 93%. This achievement represents a remarkable success in our endeavors.

$$P = \left[PAAC_1, PAAC_2, \dots, PAAC_{20}, PAAC_{20+1}, \dots, PAAC_{20+\lambda}\right]^T \tag{3}$$

$$PAAC_u = \frac{f_u}{\sum_{i=1}^{20} f_i + w \sum_{k=1}^{\lambda} T_k} \quad (1 \leq u \leq 20) \tag{4}$$

$$PAAC_u = \frac{W_T(u-20)}{\sum_{i=1}^{20} f_i + \zeta \sum_{k=1}^{\lambda} T_k} \quad (20 + 1 \leq u \leq 20 + \lambda) \tag{5}$$

Specifically, we depict the changes in data distribution before and after outlier removal. Additionally, we conducted data augmentation on the processed dataset to further enhance its accuracy.

**Di-peptide composition.** The letters AA, AC, AD, YV, YW, and YY denote protein sequences with dipeptide characteristics. There are 400 components in these sequences. The DC feature of each component is determined as follows:

$$DC(i) = \frac{DC \text{ Total } (i)}{400} \tag{6}$$

where $DC(i)$ represents the structure of $i^{th}$ dipeptide for $i = 1, 2, \dots, 400$. In vector form, this feature space is represented as:

$X_{DC} = \left[DC_{AA}, DC_{AC}, DC_{AD}, \dots, DC_{YY}\right]^T$

The di-peptide composition technique extracts features from our data, resulting in 400 dimensions or four hundred features. However, it became evident that not all these features were essential. By applying data science methods and feature engineering, it is concluded that only 229 features out of the initial 400 were necessary. Surprisingly, after this selection process, the accuracy of our results significantly improved, reaching an impressive 91% to 93%. This outcome marks a great success. The graphs illustrate the impact of outlier removal on the dataset, both before and after the process.

**Data augmentation.** The Data augmentation process is initiated by segregating our dataset into positive and negative segments. The method entails isolating patients who have

tested positive from those with negative results. Subsequently, a series of operations are designed to generate numerical replicas of the existing data, thereby augmenting the sample size. This augmentation enhances the machine learning algorithm's training procedure, attributed to the increased abundance of available data. However, it is important to note that the data transforms during the creation of these numerical duplicates, transitioning from its initial format into a list structure.

Consequently, the modified data is transited from this list format into a data frame. This procedural sequence ultimately leads to reintegrating the transformed data, thereby completing the data augmentation process.

## Development of individual classifiers

### Support vector machine

SVM classifier by creating a hyperplane with the greatest distance between any two points in the data [27,28,56]. SVM's decision surface is as follows:

$$Y(X) = \sum_{i=1}^{n} \alpha_i t_i X_i^T X + bias \tag{7}$$

We selected the parameters such as, Kernel = "rbf", Degree = 8, C = 10000, gamma = 100000, probability = True.

### Random forest

This method generates a substantial quantity of decision trees that are combined to arrive at a final decision. For training, we selected 129,361, and for testing, 86,228 samples were selected, and we came up with the best number of estimators, i.e., n = 50. In the case of dipeptide composition, we selected 2536 for training and 845 for testing, and n = 150 estimators gave optimal results.

$$Y(X) = \sum_{i=1}^{n_t} h_i(X) \tag{8}$$

### K-Nearest Neighbor (KNN)

The KNN algorithm is learned by observing samples [29,30]. Instance-based classifiers assume that the classification of unknown instances can be accomplished by comparing the unidentified instance to a known instance using a distance/similarity function [31–33,56]. The calculation of the Euclidean distance (below, denoted as d($K_i$, $K_j$), between two m-dimensional vectors $K_i$ and $K_j$ is as follows:

$$d(K_i, K_j) = \sqrt{(k_{i,1} - k_{j,1})^2 + (k_{i,2} - k_{j,2})^2 + ... + (k_{i,m} - k_{j,m})^2} \tag{9}$$

### Naïve Bayes

Bayes rules represent this learning procedure based on the notion of independent attributes/features [57–59]. The Gaussian function to train the model with equal prior

probabilities is in the following manner:

$$P(X_{f1}, X_{f2}, \ldots, X_{fn}|c) = \prod_{i=1}^{n} P(X_{fi}|c) \tag{10}$$

$$P(X_{fi}|c) = \frac{P(c_i|X_f)P(X_f)}{P(c_i)} \tag{11}$$

## XGBoost

Gradient boosting is a boosting approach that significantly lowers errors by adding several classifiers to pre-existing models. The term "gradient boosting" refers to using a gradient descent strategy to minimize loss. The steps involved in gradient boosting are as follows:

$$F_0(x) = \gamma \arg\min \sum_{i=1}^{n} L(y, \gamma) \tag{12}$$

$$\text{rim} = -\alpha \left[ \frac{\partial L(y_i, F(x_i))}{\partial F(x_i)} \right] \tag{13}$$

## Logistic regression

In categorical binary classification, a statistical machine-learning approach called logistic regression is employed [34]. The parameters we selected were C = 10, tol = 0.1, and penalty = L2.

$$P(y = 1|X) = \frac{1}{1 + e^{-\beta^T X}} \tag{14}$$

## Results and discussion

### Results on pseudo amino acid composition (Pse-AAC) data

The findings of the matrices employed in the project, including Accuracy score, F1-score, Recall [35], and Specificity respectively on the data of Pse-AAC, are displayed in Table 1 below.

Table 2 presents the results of each machine learning (ML) model concerning the data utilized, specifically the Pse-AAC data. It also includes the outcomes of additional metrics used in the research, namely Specificity and Confusion Matrix. These metrics provide insights into the True Positive, True Negative, False Positive, and False Negative values, contributing to a comprehensive evaluation of the models' performance.

**Table 1. Results on pseudo amino acid composition (Pse-AAC) data.**

| Name of Algorithm | Accuracy | F1-Score | Recall | Specificity |
|---|---|---|---|---|
| Support Vector Classifier | 92–94% | 91–92% | 91–93% | 92–94% |
| Extreme Gradient Boost | 79–85% | 63–70% | 51–55% | 92–94% |
| Logistic Regression | 66–69% | 10–20% | 6–10% | 97–98% |
| Decision Tree | 81–84% | 73–76% | 74–76% | 84–86% |
| Random Forest | 87–91% | 85–87% | 80–83% | 96–97% |
| K Nearest Neighbor | 82–86% | 72–74% | 61–64% | 93–95% |

**Table 2. Confusion matrix (Pse-AAC data).**

| Name of Algorithms | Confusion Matrix | |
|---|---|---|
| Support Vector Classifier | TN = 424 | FP = 28 |
| | FN = 14 | TN = 211 |
| Extreme Gradient Boost | TN = 26159 | FP = 2271 |
| | FN = 3435 | TP = 10890 |
| Logistic Regression | TN = 25817 | FP = 2849 |
| | FN = 11010 | TP = 3445 |
| Decision Tree | TN = 24388 | FP = 4278 |
| | FN = 3803 | TP = 10652 |
| Random Forest | TN = 28014 | FP = 808 |
| | FN = 2753 | TP = 11546 |
| K Nearest Neighbor | TN = 419 | FP = 23 |
| | FN = 95 | TP = 140 |

TN = True Negative, FP = False Positive, FN = False Negative, TP = True Positive

## Accuracy results on amino acid composition (AAC) data

The research employs Accuracy score, F1-score, Recall score, and Specificity as metrics on the AAC data. The outcomes of these metrics are presented in Table 3 below.

The following table (Table 4) presents the results of each machine learning (ML) model concerning the utilized data, namely AAC. Additionally, it showcases the outcomes of other metrics employed in the project, such as the Specificity and Confusion Matrix. These matrices provide essential values, including True Positive, True Negative, False Positive, and False Negative, contributing to a comprehensive assessment of the models' performance.

**Table 3. Result on amino acid composition (AAC) data.**

| Name of Algorithm | Accuracy | F1-Score | Recall | Specificity |
|---|---|---|---|---|
| Support Vector Classifier | 54.95% | 14.3% | 0.7% | 100% |
| Extreme Gradient Boost | 56.8% | 52.9% | 45.9% | 69% |
| Logistic Regression | 51.1% | 27.6% | 19.1% | 81.7% |
| Decision Tree | 54.4% | 52.25% | 52.9% | 55.8% |
| Random Forest | 50.6% | 41.1% | 35.4% | 64.9% |
| K Nearest Neighbor | 54.2% | 54.8% | 57% | 51% |

**Table 4. Confusion matrix (AAC data).**

| Name of Algorithms | Confusion Matrix | |
|---|---|---|
| Support Vector Classifier | TN = 271 | FP = 0 |
| | FN = 121 | TP = 62 |
| Extreme Gradient Boost | TN = 409 | FP = 23 |
| | FN = 119 | TP = 103 |
| Logistic Regression | TN = 9028 | FP = 2022 |
| | FN = 8519 | TP = 2025 |
| Decision Tree | TN = 124 | FP = 98 |
| | FN = 95 | TP = 107 |
| Random Forest | TN = 12612 | FP = 6817 |
| | FN = 11832 | TP = 6510 |
| K Nearest Neighbor | TN = 112 | FP = 105 |
| | FN = 89 | TP = 118 |

TN = True Negative, FP = False Positive, FN = False Negative, TP = True Positive

## Accuracy results on di-peptide composition (DPC)

The table below (Table 5) displays the Accuracy score, F1-score, and Recall score matrices utilized in the research and their respective outcomes when applied to the DPC data.

The performance of each machine learning model is analyzed concerning the DPC data utilized. Additionally, the Specificity and Confusion Matrix results are presented (Table 6). This matrix provides essential values such as True Positive, True Negative, False Positive, and False Negative, contributing to a comprehensive evaluation of the models' performance.

## Machine learning based dashboard

In Figures, we provide an overview of the dashboard developed using Streamlit, which is accessible through Streamlit Cloud. This interactive dashboard enables users to select their preferred model Fig 4 for analysis. Within this user-friendly interface, individuals are prompted to upload patient records directly through the web application and select a specific prediction model. Subsequently, users can review the results Fig 5 to ascertain whether an individual is affected by leukemia. Users can effortlessly select

**Table 5. Results on di-peptide composition (DPC) data.**

| Name of Algorithm | Accuracy | F1-Score | Recall | Specificity |
|---|---|---|---|---|
| Support Vector Classifier | 92–94% | 87–88% | 91–93% | 90–93% |
| Extreme Gradient Boost | 79–84% | 66–68% | 55–57% | 92–94% |
| Logistic Regression | 66–69% | 0–0% | 6–10% | 100% |
| Decision Tree | 81–84% | 70–73% | 56–59% | 96–97% |
| Random Forest | 82–84% | 67–68% | 57–58% | 94–95% |
| K Nearest Neighbor | 72–73% | 31–32% | 20–21% | 95–97% |

**Table 6. Confusion matrix (DPC data).**

| Name of Algorithms | Confusion Matrix | |
|---|---|---|
| Support Vector Classifier | TN = 416 | FP = 37 |
| | FN = 17 | TP = 207 |
| Extreme Gradient Boost | TN = 413 | FP = 25 |
| | FN = 105 | TP = 134 |
| Logistic Regression | TN = 453 | FP = 0 |
| | FN = 224 | TP = 0 |
| Decision Tree | TN = 433 | FP = 16 |
| | FN = 54 | TP = 134 |
| Random Forest | TN = 437 | FP = 23 |
| | FN = 93 | TP = 124 |
| K Nearest Neighbor | TN = 438 | FP = 15 |
| | FN = 179 | TP = 45 |

TN = True Negative, FP = False Positive, FN = False Negative, TP = True Positive

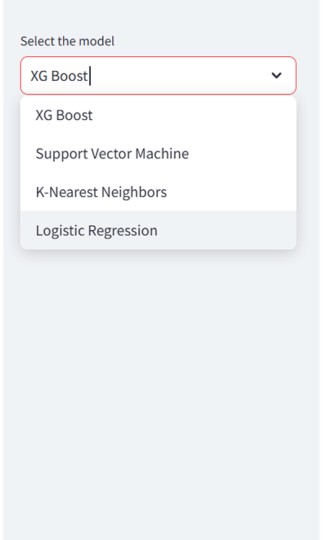

**Fig 4. Dashboard for CML overview.**

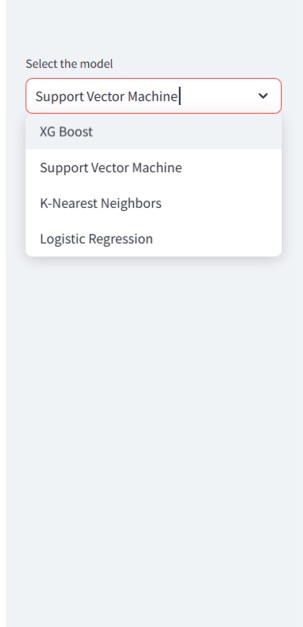

**Fig 5. Dashboard for CML with prediction.**

## Conclusion

This research is focused on Chronic Myeloid Leukemia (CML), a condition characterized by genetic mutations leading to abnormal proliferation of white blood cells, red blood cells, and platelets. While MRI and CT scans have been extensively used in cancer detection, research on protein sequence data in this domain is limited. By leveraging information from mutated genes like BCL2, HSP90, PARP, and RB, the research aims to revolutionize early CML prediction. Through rigorous data preprocessing and feature extraction techniques, we achieved an impressive accuracy rate of 92–94%. The proposed approach integrates diverse machine learning algorithms such as SVM, Decision Trees, XGBoost, Random Forest, and KNN, each offering unique strengths in pattern recognition and prediction. The resulting dashboard facilitates easy prediction of CML in patients, enhancing clinical workflows and potentially saving lives. This study sheds light on critical scientific challenges in CML research, offering insights into disease mechanisms and biomarker identification. We envision expanding this research to encompass multi-cancer detection, integrating AI and bioinformatics with healthcare systems for enhanced cancer diagnosis and improved patient outcomes.

## Acknowledgments

The authors extend their appreciation to the Princess Nourah bint Abdulrahman University Researchers Supporting Project number (PNURSP2025R513), Princess Nourah bint Abdulrahman University, Riyadh, Saudi Arabia and would like to express their gratitude to anonymous referees for their insightful comments and recommendations, which have significantly enhanced this paper. Furthermore, the authors would like to express their gratitude to Datamatics Technologies for their invaluable contributions.

## Author contributions

**Conceptualization:** Waqar Ahmad, Tahani Jaser Alahmadi.

**Data curation:** Waqar Ahmad, Abdul Raheem Shahzad, Saddam Hussain Khan.

**Formal analysis:** Abdul Raheem Shahzad, Muhammad Awais Amin, Waqas Haider Bangyal, Tahani Jaser Alahmadi, Saddam Hussain Khan.

**Funding acquisition:** Tahani Jaser Alahmadi, Saddam Hussain Khan.

**Investigation:** Muhammad Awais Amin, Waqas Haider Bangyal, Saddam Hussain Khan.

**Methodology:** Waqar Ahmad, Abdul Raheem Shahzad, Muhammad Awais Amin.

**Project administration:** Waqas Haider Bangyal, Tahani Jaser Alahmadi.

**Resources:** Tahani Jaser Alahmadi.

**Software:** Waqar Ahmad, Abdul Raheem Shahzad, Muhammad Awais Amin.

**Supervision:** Waqas Haider Bangyal, Saddam Hussain Khan.

**Validation:** Waqas Haider Bangyal, Tahani Jaser Alahmadi, Saddam Hussain Khan.

**Visualization:** Abdul Raheem Shahzad, Muhammad Awais Amin.

**Writing – original draft:** Muhammad Awais Amin.

**Writing – review & editing:** Waqar Ahmad, Muhammad Awais Amin, Saddam Hussain Khan.

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
