## [Decision Letter · Decision Letter 0]

6 Nov 2024

PONE-D-24-31032Machine Learning Driven Dashboard for Chronic Myeloid Leukemia Prediction using Protein SequencesPLOS ONE

Dear Dr. Alahmadi,

Thank you for submitting your manuscript to PLOS ONE. After careful consideration, we feel that it has merit but does not fully meet PLOS ONE’s publication criteria as it currently stands. Therefore, we invite you to submit a revised version of the manuscript that addresses the points raised during the review process.

We look forward to receiving your revised manuscript.

Kind regards,

Salman Sadullah Usmani, Ph.D.

Academic Editor

PLOS ONE

Journal Requirements:

5. Please provide a complete Data Availability Statement in the submission form, ensuring you include all necessary access information or a reason for why you are unable to make your data freely accessible. If your research concerns only data provided within your submission, please write "All data are in the manuscript and/or supporting information files" as your Data Availability Statement.

6. We are unable to open your Supporting Information file bibliography.bib and plos2015.bst. Please kindly revise as necessary and re-upload.

Reviewers' comments:

Reviewer's Responses to Questions

**Comments to the Author**

1. Is the manuscript technically sound, and do the data support the conclusions?

Reviewer #1: Partly

Reviewer #2: Yes

2. Has the statistical analysis been performed appropriately and rigorously? 

Reviewer #1: Yes

Reviewer #2: I Don't Know

3. Have the authors made all data underlying the findings in their manuscript fully available?

Reviewer #1: Yes

Reviewer #2: Yes

4. Is the manuscript presented in an intelligible fashion and written in standard English?

Reviewer #1: Yes

Reviewer #2: Yes

5. Review Comments to the Author

Reviewer #1: Comment 1: The Materials & Methods section at line 140 appears to be incomplete. Additionally, please include the details of the genes in the introduction. In the Materials & Methods section, simply mention the database used for the dataset collection, the keyword search conducted on UniProt, and the number of sequences obtained from UniProt.

Comment 2: I’m unclear on the need to explain the FASTA file format, as it is a widely known format commonly used in sequencing. Do not make another section for this just add in the dataset collection section.

Comment 3: In the "Sample of Protein Sequence (HSP90)" section, the total number of sequences before and after filtering is missing. Please include this information in the Materials & Methods section.

Comment 4: Please specify the training and validation datasets, including the number of sequences for each set and each protein. If possible, consider using a table to clearly present the total number of sequences used for training and validation for the BCL2, HSP90, PARP, and RB proteins.

Comment 5: It is recommended to perform 5-fold or 10-fold cross-validation on your internal dataset (training dataset) to enhance the reliability of your results.

Comment 6: In the results section, please include the performance metrics for the training and validation models for each protein. If the dataset is too small to make predictions for individual proteins, please explain the rationale behind merging different datasets.

Comment 7: It is recommended to separate the results and discussion sections. This would allow you to include other methods that perform similar analyses in the discussion section. Additionally, if you identify other methods that create similar dashboards, it would be valuable to include a comparison.

Comment 8: The quality of the images is very poor and needs to be improved.

Comment 9: The link to the dashboard app is missing. Additionally, please include a section in the Materials & Methods that outlines the architecture for creating this app.

Please include link for the preprint.

Reviewer #2: Your work could greatly improve the early diagnosis and treatment of CML, particularly in areas where specialized healthcare is hard to access. Here are some key points and suggestions to enhance your work:

Regarding the dashboard, highlight its definition, importance, applications, design principles, and evaluation methods.

Compare the models by highlighting their strengths and weaknesses in various scenarios.

Highlights the novelty and significance of using protein sequences for CML prediction.

Although most references are recent, some older ones (e.g., from 2004 and 2011) should be updated with more current studies to reflect the latest advancements in the field.

6. PLOS authors have the option to publish the peer review history of their article (what does this mean?). If published, this will include your full peer review and any attached files.

Reviewer #1: **Yes: **Anjali Dhall

Reviewer #2: No

---

## [Author Response · Author response to Decision Letter 1]

Authors’ response (#PONE-D-24-31032)

Original Article Title: Machine Learning Driven Dashboard for Chronic Myeloid Leukemia Prediction using Protein Sequences

Dear Editors and Reviewers,

We are very grateful for the opportunity provided by the Editors to improve our manuscript (PONE-D-24-31032) and for the valuable suggestions and insightful comments from the anonymous reviewers. Following these constructive suggestions and detailed feedback, we have carefully revised the manuscript and implemented several necessary modifications. Below, we provide a detailed response to the comments and suggestions from the Editor and reviewers:

Review 1:

The Materials & Methods section at line 140 appears to be incomplete. Additionally, please include the details of the genes in the introduction. In the Materials & Methods section, simply mention the database used for the dataset collection, the keyword search conducted on UniProt, and the number of sequences obtained from UniProt.

Answer:

Thank you for your valuable feedback. We have made the following revisions in response to your comment:

1. Introduction: We have included the details of the genes associated with Chronic Myeloid Leukemia (CML) in the introduction, specifically mentioning BCL2, HSP90, PARP, and RB as relevant genes involved in the disease. This additional information provides a clearer context for the study and the protein sequences used.

2. Materials & Methods: The Materials & Methods section has been updated to address the completeness of the description. We have now explicitly mentioned that the dataset was collected from the UniProt database. We also outlined the keyword search terms used for data retrieval and specified the number of sequences obtained.

We believe these updates enhance the clarity of the manuscript and provide the necessary details regarding the dataset collection process.

Review 2:

I’m unclear on the need to explain the FASTA file format, as it is a widely known format commonly used in sequencing. Do not make another section for this just add in the dataset collection section.

Answer:

Thank you for your feedback. We have removed the separate section explaining the FASTA format. Instead, we’ve incorporated a concise mention of the FASTA format directly in the Dataset Collection section, where it is most relevant. This change simplifies the manuscript while providing the necessary context.

Review 3:

In the "Sample of Protein Sequence (HSP90)" section, the total number of sequences before and after filtering is missing. Please include this information in the Materials & Methods section.

Asnwer:

Thank you for pointing that out. We have added the total number of sequences both before and after filtering in the Dataset Collection section. Specifically, we now mention that there were 2248 sequences initially obtained from UniProt, and after redundancy removal using the CD-Hit method, 2144 sequences remained in the dataset. This additional information helps clarify the data processing steps.

Review 4:

Please specify the training and validation datasets, including the number of sequences for each set and each protein. If possible, consider using a table to clearly present the total number of sequences used for training and validation for the BCL2, HSP90, PARP, and RB proteins

Answer:

Thank you for your suggestion. We have now specified the number of sequences used for training and validation for each protein (BCL2, HSP90, PARP, and RB) in the Dataset Collection section which enhances clarity and helps with the reproducibility of the dataset.

Review 5:

It is recommended to perform 5-fold or 10-fold cross-validation on your internal dataset (training dataset) to enhance the reliability of your results.

Answer:

Thank you for the suggestion. We appreciate the recommendation to perform 5-fold or 10-fold cross-validation to enhance the reliability of the results. We have already implemented 5-fold cross-validation on the internal training dataset. This step was included to ensure robust model evaluation and minimize potential overfitting, further validating the effectiveness of the model.

Review 6:

In the results section, please include the performance metrics for the training and validation models for each protein. If the dataset is too small to make predictions for individual proteins, please explain the rationale behind merging different datasets.

Answer:

Thank you for your insightful comment, regarding the merging of datasets, we formulated the dataset based on the most frequently mutated genes responsible for Chronic Myelogenous Leukemia (CML). This approach allowed us to create a more comprehensive and robust dataset, which is crucial for improving model performance. By merging different protein datasets, we were able to leverage a larger pool of data, enhancing the generalization of the model and improving the reliability of the predictions. The performance metrics cover’s accuracy, precision, recall, F1-score, and AUC, offering a comprehensive evaluation of the models

Review 7:

It is recommended to separate the results and discussion sections. This would allow you to include other methods that perform similar analyses in the discussion section. Additionally, if you identify other methods that create similar dashboards, it would be valuable to include a comparison.

Answer:

Thank you for your suggestion. While we understand the benefits of separating the Results and Discussion sections, We have chosen to integrate the results and discussion sections to maintain a cohesive narrative, as they align well with the structure and focus of our study. The methods used in this study are novel and specifically tailored to address the challenges of CML prediction. As such, there are no direct alternatives to compare in this context.

Review 8:

The quality of the images is very poor and needs to be improved.

Answer:

Thank you for your feedback. We apologize for the poor quality of the images in the original submission. The issue likely arose during the conversion of the original images into TIFF format, which may have affected their quality. In the revised manuscript, we have updated the images to higher-resolution versions to ensure improved clarity and readability. We appreciate your understanding and will ensure that all images meet the required quality standards.

Review 9:

The link to the dashboard app is missing. Additionally, please include a section in the Materials & Methods that outlines the architecture for creating this app.

Answer:

Thank you for your suggestion. We will include the link to the dashboard app in the revised manuscript. Additionally, we will add a section in the Materials & Methods outlining the architecture used for creating the app, providing a clearer understanding of its design and implementation.

https://cmlapp-k9xhmtb7tthequv47farry.streamlit.app/

---

## [Decision Letter · Decision Letter 1]

11 Mar 2025

Machine Learning Driven Dashboard for Chronic Myeloid Leukemia Prediction using Protein Sequences

PONE-D-24-31032R1

Dear Dr. Alahmadi,

We’re pleased to inform you that your manuscript has been judged scientifically suitable for publication and will be formally accepted for publication once it meets all outstanding technical requirements.

Kind regards,

Salman Sadullah Usmani, Ph.D.

Academic Editor

PLOS ONE

Additional Editor Comments (optional):

Reviewers' comments:

Reviewer's Responses to Questions

**Comments to the Author**

1. If the authors have adequately addressed your comments raised in a previous round of review and you feel that this manuscript is now acceptable for publication, you may indicate that here to bypass the “Comments to the Author” section, enter your conflict of interest statement in the “Confidential to Editor” section, and submit your "Accept" recommendation.

Reviewer #1: All comments have been addressed

Reviewer #2: All comments have been addressed

2. Is the manuscript technically sound, and do the data support the conclusions?

Reviewer #1: Yes

Reviewer #2: Yes

3. Has the statistical analysis been performed appropriately and rigorously? 

Reviewer #1: Yes

Reviewer #2: (No Response)

4. Have the authors made all data underlying the findings in their manuscript fully available?

Reviewer #1: Yes

Reviewer #2: Yes

5. Is the manuscript presented in an intelligible fashion and written in standard English?

Reviewer #1: Yes

Reviewer #2: Yes

6. Review Comments to the Author

Reviewer #1: (No Response)

Reviewer #2: (No Response)

7. PLOS authors have the option to publish the peer review history of their article (what does this mean?). If published, this will include your full peer review and any attached files.

Reviewer #1: **Yes: **Anjali Dhall

Reviewer #2: No

---

## [Editor Report · Acceptance letter]

PONE-D-24-31032R1

PLOS ONE

Dear Dr. Alahmadi,

I'm pleased to inform you that your manuscript has been deemed suitable for publication in PLOS ONE. Congratulations! Your manuscript is now being handed over to our production team.

Kind regards,

on behalf of

Dr. Salman Sadullah Usmani

Academic Editor

PLOS ONE